



Atmospheric
Measurement
Techniques

# First validation of Aeolus wind observations by airborne Doppler wind lidar measurements

**Benjamin Witschas[1], Christian Lemmerz[1], Alexander Geiß[2], Oliver Lux[1], Uwe Marksteiner[1], Stephan Rahm[1], Oliver Reitebuch[1], and Fabian Weiler[1]**

[1]Deutsches Zentrum für Luft- und Raumfahrt e.V. (DLR), Institut für Physik der Atmosphäre,
82234 Oberpfaffenhofen, Germany
[2]Meteorologisches Institut, Ludwig-Maximilians-Universität, 80333 Munich, Germany

**Correspondence:** Benjamin Witschas (benjamin.witschas@dlr.de)

**Abstract.** Soon after the launch of Aeolus on 22 August 2018, the first ever wind lidar in space developed by the European Space Agency (ESA) has been providing profiles of the component of the wind vector along the instrument's line of sight (LOS) on a global scale. In order to validate the quality of Aeolus wind observations, the German Aerospace Center (Deutsches Zentrum für Luft- und Raumfahrt e.V., DLR) recently performed two airborne campaigns over central Europe deploying two different Doppler wind lidars (DWLs) on board the DLR Falcon aircraft. The first campaign – WindVal III – was conducted from 5 November 2018 until 5 December 2018 and thus still within the commissioning phase of the Aeolus mission. The second campaign – AVATARE (Aeolus Validation Through Airborne Lidars in Europe) – was performed from 6 May 2019 until 6 June 2019. Both campaigns were flown out of the DLR site in Oberpfaffenhofen, Germany, during the evening hours for probing the ascending orbits. All together, 10 satellite underflights with 19 flight legs covering more than 7500 km of Aeolus swaths were performed and used to validate the early-stage wind data product of Aeolus by means of collocated airborne wind lidar observations for the first time. For both campaign data sets, the statistical comparison of Aeolus horizontal line-of-sight (HLOS) observations and the corresponding wind observations of the reference lidar (2 µm DWL) on board the Falcon aircraft shows enhanced systematic and random errors compared with the bias and precision requirements defined for Aeolus. In particular, the systematic errors are determined to be 2.1 m s$^{-1}$ (Rayleigh) and 2.3 m s$^{-1}$ (Mie) for WindVal III and −4.6 m s$^{-1}$ (Rayleigh) and −0.2 m s$^{-1}$ (Mie) for AVATARE. The corresponding random errors are determined to be 3.9 m s$^{-1}$ (Rayleigh) and 2.0 m s$^{-1}$ (Mie) for WindVal III and 4.3 m s$^{-1}$ (Rayleigh) and 2.0 m s$^{-1}$ (Mie) for AVATARE. The Aeolus observations used here were acquired in an altitude range up to 10 km and have mainly a vertical resolution of 1 km (Rayleigh) and 0.5 to 1.0 km (Mie) and a horizontal resolution of 90 km (Rayleigh) and down to 10 km (Mie). Potential reasons for those errors are analyzed and discussed.

## 1 Introduction

Since 22 August 2018, the first European spaceborne lidar and the first ever spaceborne Doppler wind lidar, Aeolus, developed by ESA has been circling in its sun-synchronous orbit at about 320 km altitude (ESA, 1999). Aeolus is carrying a single payload, namely the Atmospheric Laser Doppler Instrument (ALADIN), which provides profiles of the component of the wind vector along the instrument's LOS direction on a global scale from the ground up to about 30 km in the stratosphere (ESA, 1999; Stoffelen et al., 2005; Reitebuch, 2012; Kanitz et al., 2019). With that, the Aeolus mission is primarily aiming to improve numerical weather prediction (NWP) and medium-range weather forecast (e.g., Weissmann and Cardinali, 2007; Tan et al., 2007; Marseille et al., 2008; Horányi et al., 2015).

ALADIN is a direct detection wind lidar operating at a laser wavelength of 354.8 nm and is able to retrieve LOS

wind speeds by exploiting the Doppler shift of light backscattered from molecules and from particles. In order to do so, ALADIN is equipped with two different frequency discriminators, namely a Fizeau interferometer that is used to analyze the frequency shift of the narrowband particulate return signal by means of the so-called fringe imaging technique (McKay, 2002) and two coupled Fabry–Pérot interferometers that are used to analyze the frequency shift of the broadband molecular return signal by the so-called double-edge technique (e.g., Chanin et al., 1989; Flesia and Korb, 1999). This high-spectral-resolution receiver configuration also provides the possibility to retrieve information on the vertical distribution of aerosol and cloud optical properties such as backscatter and extinction coefficients (Ansmann et al., 2007; Flamant et al., 2008).

The direct detection measurement principle requires regular instrument calibration, a stable instrument alignment and further post-processing that relates the measured signal levels to a frequency Doppler shift which can then be translated into a wind speed (Dabas et al., 2008; Lux et al., 2018; Marksteiner et al., 2018; Zhai et al., 2020). Hence, in particular the accuracy of wind speeds retrieved from direct detection wind lidars strongly depends on the aforementioned points. This also means that a validation of Aeolus winds by means of independent ground-based and airborne measurements is inevitable. For that reason, ESA already provided preliminary Aeolus data in a very early stage of the mission (since 16 December 2018) to approved cal–val (calibration and validation) teams that especially perform ground-based and airborne measurements for validation purposes (https://aeolus-ds.eo.esa.int/oads/access/, last access: 12 November 2019).

As one of these teams, DLR recently performed two airborne campaigns over central Europe, namely the WindVal III campaign and the AVATARE campaign with the DLR Falcon research aircraft equipped with two wind lidar systems that have been deployed in several Aeolus pre-launch campaigns within the last 10 years (Marksteiner, 2013; Marksteiner et al., 2018; Schäfler et al., 2018; Lux et al., 2018). During both campaigns, 10 satellite underflights covering more than 7500 km of Aeolus swaths were acquired. Based on these measurements, this paper presents the first validation of the early-stage Aeolus horizontal-line-of-sight (HLOS) wind product (Level 2B). In particular, the Aeolus data are compared to 2 μm DWL measurements which act as a reference due to their low systematic and random errors that come along with the coherent measurement principle of the system. A study of the Aeolus measurement principle, its calibration procedures and retrieval algorithms is performed based on ALADIN airborne demonstrator (A2D) observations as discussed in Lux et al. (2020a).

First, an overview of the two validation campaigns is given, followed by a discussion of the ALADIN and 2 μm DWL instrumental setup and measurement schemes. Afterwards, the procedure of matching the different reso-

lutions of the used data sets is explained and a statistical comparison is performed. Finally, potential reasons for the observed enhanced systematic and random errors of Aeolus winds are discussed.

## 2 Validation campaign overview

Still within the commissioning phase of Aeolus, DLR performed a first airborne Aeolus validation campaign (WindVal III) from the site in Oberpfaffenhofen, Germany, in the timeframe from 5 November 2019 to 5 December 2018. Half a year later, a second airborne Aeolus validation campaign called AVATARE was conducted from 6 May until 6 June 2019.

During both campaigns, the DLR Falcon was equipped with two wind lidar systems that have been deployed in several Aeolus pre-launch campaigns such as WindVal I (Marksteiner et al., 2018) and WindVal II (Schäfler et al., 2018; Lux et al., 2018), both flown out of Keflavík, Iceland. In particular, the Falcon hosted the A2D, which is a prototype of the ALADIN instrument with representative design and measurement principle (Reitebuch et al., 2009). The A2D was developed by the former European Aeronautic Defence and Space Company (EADS-Astrium – now Airbus Defence and Space) together with DLR in order to validate the ALADIN measurement principle, calibration procedures, retrieval algorithms and wind product quality before and after the launch of Aeolus. Additionally, a coherent detection wind lidar (2 μm DWL) with a high sensitivity to particulate returns was flown and acted as a reference system (Witschas et al., 2017)

Whereas the flights performed during WindVal I and WindVal II resulted in refinements of the Aeolus wind retrieval algorithms based on measurements performed in real atmosphere, wind observations collocated with Aeolus could be acquired during WindVal III and AVATARE, enabling the first ever validation of the early-stage Aeolus HLOS winds (Level 2B). In order to do so, four satellite underflights composed of eight flight legs were conducted during WindVal III over central Europe, covering more than 3000 km of Aeolus swaths. During AVATARE, six satellite underflights composed of 11 flight legs were performed over central Europe covering more than 4500 km along the Aeolus swath. Thus, data of 19 flight legs from 10 satellite underflights that cover more than 7500 km of Aeolus swaths are available and used for the validation Aeolus HLOS winds. An overview of the flight tracks flown during WindVal III and AVATARE is shown in Fig. 1, left and right, respectively. Further details about the flight times of the Falcon aircraft and the overflight times of Aeolus are given in Table 1.

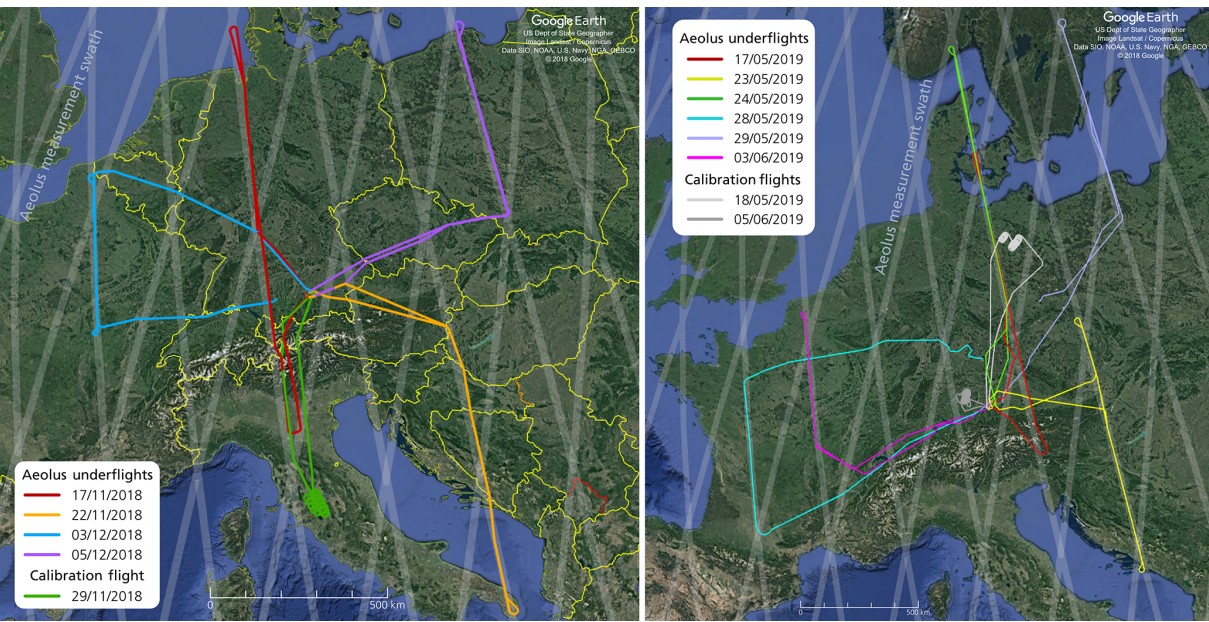

**Figure 1.** Flight tracks of the Falcon aircraft during the WindVal III campaign performed from 17 November to 5 December 2018 (left) and the AVATARE campaign performed from 17 May to 6 June 2019 (right). Each color represents a single flight. The Aeolus measurement swath is shown in gray. During the probed evening satellite tracks, the Aeolus moving direction was always from south to north (ascending orbit).

**Table 1.** Overview of Aeolus underflights performed during the WindVal III and the AVATARE campaign.

| | | Falcon flight | | Aeolus underflight | |
|---|---|---|---|---|---|
| | Date | Time (UTC) | Route | Start and stop time (UTC) | Geolocation |
| WindVal III | 17 November 2018 | 15:14 to 19:14 | OBF to OBF | 17:01:21 to 17:03:56 | 44.7° N, 10.6° E to 54.9° N, 7.8° E |
| | 22 November 2018 | 14:29 to 17:56 | OBF to OBF | 16:34:14 to 16:36:02 | 40.0° N, 18.3° E to 47.2° N, 16.5° E |
| | 3 December 2018 | 15:48 to 19:31 | FMM to OBF | 17:27:55 to 17:28:51 | 47.1° N, 3.6° E to 50.8° N, 2.6° E |
| | 3 December 2018 | 14:56 to 18:22 | OBF to OBF | 16:23:50 to 16:25:02 | 50.2° N, 19.0° E to 54.9° N, 17.5° E |
| AVATARE | 17 May 2019 | 15:36 to 18:46 | OBF to OBF | 16:48:39 to 16:51:01 | 46.3° N, 13.4° E to 55.5° N, 10.7° E |
| | 23 May 2019 | 14:30 to 18:08 | OBF to OBF | 16:34:55 to 16:36:55 | 42.9° N, 17.5° E to 50.5° N, 15.6° E |
| | 24 May 2019 | 15:28 to 19:09 | OBF to OBF | 16:50:01 to 16:52:18 | 51.2° N, 12.2° E to 59.0° N, 9.4° E |
| | 28 May 2019 | 15:54 to 19:13 | NUE to OBF | 17:40:05 to 17:41:10 | 44.0° N, 1.1° E to 48.2° N, 0.1° E |
| | 29 May 2019 | 15:26 to 19:11 | OBF to OBF | 16:24:40 to 16:26:12 | 53.5° N, 18.1° E to 59.4° N, 15.9° E |
| | 3 June 2019 | 15:26 to 18:46 | OBF to OBF | 17:27:50 to 17:28:48 | 46.8° N, 3.6° E to 50.6° N, 2.6° E |

The time gives the duration between takeoff and landing. The flight route is indicated by the IATA (International Air Transport Association) airport code.
OBF: Oberpfaffenhofen airport. FMM: Allgäu airport Memmingen. NUE: Nuremberg airport.

## 3 The Atmospheric Laser Doppler Instrument (ALADIN) on board Aeolus

In this section, the Aeolus satellite and its instrument ALADIN are briefly introduced, including its measurement procedure and resulting data products. For more information regarding these topics, please refer to ESA (1999), Reitebuch (2012), Reitebuch et al. (2019), Kanitz et al. (2019) and Straume et al. (2018, 2019), for example.

### 3.1 Instrument description

The Aeolus satellite was launched on 22 August 2018. It has a weight of 1360 kg and a launch configuration dimension of 4.6 m × 1.9 m × 2.0 m, and it can provide a power of 2.4 kW. It flies in a 320 km sun-synchronous orbit with an inclination of 97°, leading to a 7 d repeat cycle. Aeolus carries a single payload, ALADIN, which is a direct-detection wind lidar operating at an ultraviolet wavelength of 354.8 nm. ALADIN emits short laser pulses (≈ 40 to 70 mJ, 50.5 Hz) down to the atmosphere where a few of the photons are backscattered on air molecules, aerosols and hydrometeors. The backscattered

light is collected with a 1.5 m diameter telescope and directed to the optical receiver that is used to detect the Doppler frequency shift of the backscattered light and with that the wind velocity in the LOS direction at different altitudes. In order to do so, ALADIN is equipped with two different frequency discriminators, namely a Fizeau interferometer that is used to analyze the frequency shift of the narrowband particulate backscatter signal (Mie) and two sequentially coupled Fabry–Pérot interferometers that are used to analyze the frequency shift of the broadband molecular return signal (Rayleigh). Both the Rayleigh and Mie channels sample the backscatter signal time resolved to 24 bins with a vertical resolution between 0.25 and 2.0 km. The horizontal resolution of the wind observations is about 90 km for the Rayleigh channel and down to 10 km for the Mie channel with overall subsample information on a 3 km scale. Furthermore, due to the high-spectral-resolution receiver configuration, information on the vertical distribution of aerosol and cloud optical properties such as backscatter and extinction coefficients can also be retrieved from Aeolus data (Ansmann et al., 2007; Flamant et al., 2008).

As demonstrated by several authors (e.g., Reitebuch et al., 2018; Lux et al., 2018; Marksteiner et al., 2018; Zhai et al., 2020), the direct detection measurement principle requires regular instrument calibration and further post-processing that relates the measured signal levels to a frequency Doppler shift which can then be converted into a wind speed. Hence, the systematic error of wind speeds retrieved from direct detection wind lidars in particular strongly depends on the quality of the instrument calibration and the alignment stability of the instrument itself. Thus, in order to verify if the Aeolus instrument calibration procedures and processing steps are robust, validation measurements are inevitable.

## 3.2   Aeolus data products

The Aeolus data processing chain offers different data product levels containing different types of information. A short overview of them is given in this section. For additional information it is referred to Tan et al. (2017, ?), ESA (2016) and Rennie (2018), for example.

The Level 0 data contain the raw data of ALADIN as well as the instrument housekeeping data and the housekeeping data of the satellite platform. The assignment of the geolocation to each measurement and the full processing of the satellite housekeeping data is done in the Level 1A processor. The Level 1B data already provide processed ground echo data and preliminary HLOS wind observations that have not been corrected for atmospheric temperature and pressure (Reitebuch et al., 2018). Additionally, the viewing geometry data are available (Tan et al., 2008). The Level 2B data contain the time series of fully processed profiles of HLOS wind along the satellite orbit. It is the data product that is also used by the European Centre for Medium-Range Weather Forecasts (ECMWF) for NWP (Tan et al., 2017; Rennie, 2018)

and for the validation by means of 2 μm DWL measurements as discussed later. It is worth mentioning that the sign of the HLOS winds is defined such that it is positive for winds blowing away from the satellite. For instance, for an ascending orbit, when the satellite moves from south to north and the laser is pointing eastwards, westerly winds lead to positive HLOS winds.

Additionally, there are also Level 2C data available which contain the time series of three-dimensional wind vector profiles along the satellite track, which are produced by the ECMWF model after ingestion of Level 2B data.

From Level 1B to Level 2B, the following important steps are performed. First, the single measurements are grouped into observations. By doing so, the horizontal resolution and the noise of the respective wind observation are controlled. Furthermore, the measurements are classified by means of the optical properties of the atmosphere. In particular, the wind observations are classified into Rayleigh-clear winds, indicating wind observations in aerosol-poor atmosphere, and Mie-cloudy winds, indicating winds acquired from particulate backscatter, predominately from clouds. There are also Rayleigh-cloudy and Mie-clear winds available in the data product which are not further discussed within this study. Moreover, a temperature and pressure correction is applied for the Rayleigh-wind retrieval which is needed in order to avoid systematic errors (Dabas et al., 2008). As the Rayleigh–Brillouin spectrum of molecular scattered light depends on temperature and pressure (Witschas et al., 2010, 2014; Witschas, 2011a, b), any temperature and pressure differences between instrument response calibration and wind observation have to be taken into account. Additionally, a potential cross talk between the Mie and the Rayleigh channel is corrected within the Level 2B processor. Rayleigh-clear winds are usually retrieved for a backscatter ratio from 1.0 to 1.4, where the backscatter ratio is defined as the ratio of the total backscatter coefficient (particles and molecules) to the molecular component. Thus, for the larger scattering ratios (close to 1.4) the sensitivity of the Rayleigh channel might already be impacted by the enhanced Mie signal which has to be considered for the wind retrieval in order to avoid systematic errors. Besides these processing steps, uncertainty estimates and quality flags are calculated for each wind observation and can be used for quality control.

It is worth mentioning that the Level 2B HLOS winds used in this study are still in an early-stage state. The Level 1B and Level 2B processors are continuously updated, and particular improvements have already been performed; however, the satellite data have not been reprocessed yet. For the Level 2B HLOS winds analyzed here, one and the same instrument calibration file was used from the start of the mission until 16 May 2019. Additionally, ECMWF model comparisons from September 2018 were used to further correct a remaining systematic bias. On 16 May 2019, the calibration file was updated. Thus, the two campaigns discussed here are compared to Aeolus data processed with different instrument cal-

ibration files. Another difference between both campaigns is the resolution of Mie winds. On 5 March 2019 (08:44 UTC), the resolution of Mie winds was increased by decreasing the horizontal averaging down to about 10 km. Furthermore, the range-gate settings of Aeolus were changed on 26 February 2019 (00:00 UTC) such that they follow the ground elevation, which also increases the number of available data points due to smaller range gates in altitudes with airborne lidar measurements.

## 4 The 2 µm Doppler wind lidar at DLR

The 2 µm DWL has been operated by DLR for almost 20 years and has been deployed in several ground and airborne field campaigns for measuring aircraft wake vortices (Köpp et al., 2004), aerosol optical properties (Chouza et al., 2015, 2017), horizontal wind speeds over the Atlantic Ocean as input data for assimilation experiments (Weissmann et al., 2005; Schäfler et al., 2018), and horizontal and vertical wind speeds to study the life cycle of gravity waves (Witschas et al., 2017). In addition to that, the system was applied in several Aeolus pre-launch campaigns conducted within the last 10 years (e.g., Marksteiner et al., 2018; Lux et al., 2018).

In this section, the 2 µm DWL instrument is shortly described, followed by an explanation of the corresponding measurement procedure and wind retrieval algorithm. Afterwards, the accuracy and precision of the derived wind speeds are discussed by means of comparison to dropsonde measurements available from previous campaigns.

### 4.1 Instrument description

The 2 µm DWL is a coherent detection wind lidar system based on a Tm:LuAG laser operating at a wavelength of 2022.54 nm (vacuum), a laser pulse energy of 1 to 2 mJ and a pulse repetition rate of 500 Hz, ensuring eye-safe operation. The system was built by CLR Photonics, Inc. (today Lockheed Martin Coherent Technologies, Inc.) and has been deployed at DLR since October 1999.

The 2 µm DWL is composed of three units, namely (1) a transceiver head containing the laser, a 11 cm afocal telescope, receiver optics, detectors and a double-wedge scanner enabling us to steer the laser beam to any position within a 30° cone angle; (2) a power supply and the cooling unit of the laser, mounted in a separate rack; and (3) a rack containing the data acquisition unit and the control electronics, developed by DLR. For a more detailed description of the 2 µm DWL including a listing of the system specifications, please refer to Witschas et al. (2017).

### 4.2 Measurement procedure and wind retrieval

In order to measure the three-dimensional wind speed and direction, the velocity–azimuth display (VAD) scan technique is applied (Browning and Wexler, 1968). That is, a conical step-and-stare scan around the vertical axes with an off-nadir angle of 20° is performed for 21 LOS positions, separated by 18° in the azimuth direction. Considering a 1 s averaging time for each LOS measurement and an additional second in order to change the laser beam pointing direction, one scanner revolution takes about 42 s. By further taking into account the aircraft speed of about 200 m s$^{-1}$, the horizontal resolution of 2 µm DWL wind observations is about 8.4 km, depending on the actual ground speed of the aircraft. The vertical resolution of the wind observations is determined by the laser pulse length and the averaging interval which is set to be 100 m.

In order to retrieve wind speed and wind direction profiles from the single LOS measurements performed during one scanner revolution, several techniques are available (Smalikho, 2003). As discussed by Witschas et al. (2017), an algorithm based on a maximum function of accumulated spectra (MFAS) is used as the baseline for the 2 µm DWL. The MFAS algorithm retrieves wind speed and wind direction without estimating single LOS wind velocities and thus yields valid wind estimates even in regions of low signal-to-noise ratios (SNRs). In particular, the spectra of all 21 LOS measurements are shifted to be proportional to their azimuth angle and an assumed wind vector. Afterwards, all spectra are accumulated, and the maximum of the accumulated spectra is determined. For a correctly assumed wind vector, the accumulated spectra have a maximum and thus indicate the prevailing wind vector. By applying the MFAS algorithm to one scanner revolution, the horizontal resolution and vertical resolution of the retrieved wind vectors are about 8.4 km and 100 m, respectively.

Considering the lower resolution of Aeolus data, which is about 90 km for the Rayleigh-clear winds and down to 10 km for the Mie-cloudy winds (horizontal) and between 0.25 and 2 km (vertical), it was investigated if an increased number of averaged spectra for the MFAS algorithm could further improve the 2 µm DWL data coverage and with that increase the number of data points available for comparison to Aeolus observations. In particular, a sliding window of five scanner revolutions (90 LOS measurements) and five range gates (500 m) is used, decreasing the effective horizontal and vertical resolution of the retrieved wind vectors to 42 km and 500 m, respectively, whereas the data are still available on the one scanner revolution grid, which is 8.4 km and 100 m, respectively.

In Fig. 2, an example of the wind speed retrieved from 2 µm DWL measurements performed on the first flight leg of the first ever Aeolus underflight performed on 17 November 2018 during the WindVal III campaign is shown. The flight leg ranges from 44.85 to 54.82° N, which corresponds to a track length of 1146 km. The leg started south of the Alps at 15:57 UTC and ended in the north of Germany at 17:45 UTC (see also Fig. 1, left, red line). The Aeolus overflight was at around 17:02 UTC. The top panel indicates data processed with the MFAS algorithm for one scanner revo-

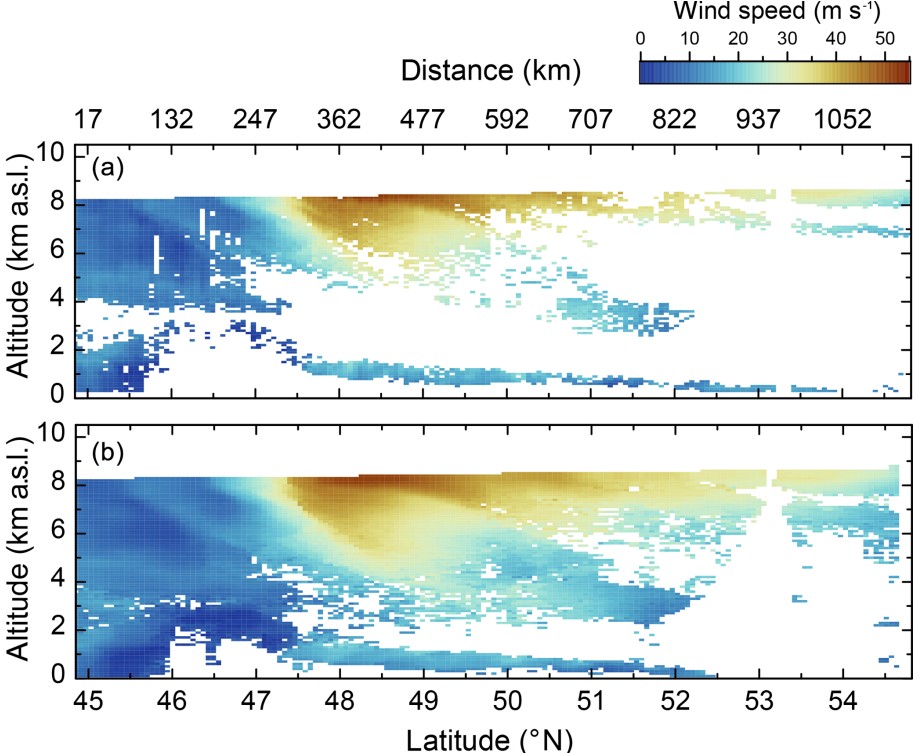

**Figure 2.** Wind speed retrieved from 2 μm DWL data by means of the MFAS algorithm for one scanner revolution **(a)** and five scanner revolutions and five range gates with a sliding window **(b)**, during the first ever Aeolus underflight performed on 17 November 2018 during the WindVal III campaign (see also Table 1 and Fig. 1, left, red line). The flight leg ranges from 44.85 to 54.82° N which corresponds to 1146 km track length. The leg started in the south at 15:57 UTC and ended in the north at 17:45 UTC. The satellite overflight was at around 17:02 UTC. White indicates areas with no valid wind measurements due to aerosol-poor atmospheric conditions and a corresponding insufficient SNR.

lution and 100 m vertical resolution; the bottom panel shows data processed with the MFAS algorithm for five scanner revolutions and 500 m vertical resolution (sliding window).

It can be seen that the data coverage for the five-scanner-revolution average is remarkably increased. In particular, the retrieval by means of one scanner revolution yields 4693 valid data points out of 12 517 data points which would give full coverage. Thus, the data coverage with one scanner revolution is about 37.5 %. On the other hand, the retrieval by means of five scanner revolutions yields 8719 valid data points which corresponds to a data coverage of 70 % and thus an increase of 86 % compared to the one scanner revolution. Apart from that, it can be seen that detailed structures, for instance in the vicinity of the jet stream (47.5 to 50.0° N), become less pronounced or blurred due to the decreased resolution of the data. However, as the resolution of the satellite data is even coarser, this should not be an issue for comparison.

In order to prove this hypothesis, the wind speeds retrieved by means of one scanner revolution ($v_{2\,\mu m_{1-scan}}$) and five scanner revolutions ($v_{2\,\mu m_{5-scans}}$) are analyzed. In particular, the difference of both data sets ($v_{2\,\mu m_{1-scan}} - v_{2\,\mu m_{5-scans}}$) for all common data points of all flights flown during the

AVATARE campaign (see also Table 1) and the corresponding mean and standard deviation (SD) is calculated. A histogram of the wind speed difference is shown in Fig. 3.

All together, more than 40 000 data points contribute to this analysis. It can be seen that the systematic error of the wind speed difference is $0.04\,\mathrm{m\,s^{-1}}$ and thus negligible for the comparison to Aeolus data. The random error (standard deviation) is determined to be $1.24\,\mathrm{m\,s^{-1}}$. Assuming that both data sets contribute equally, the random error of 2 μm DWL wind speeds can be estimated to be $\sigma_{2\,\mu m} = (\sigma_{\text{difference}}/2)^{1/2} = 0.88\,\mathrm{m\,s^{-1}}$, which is in line with previous comparisons to dropsonde measurements as shown in Sect. 4.3, Table 2.

Considering that, it was decided to use the 2 μm DWL data retrieved by means of the modified MFAS algorithm using five scanner resolutions (horizontal) and five range gates (vertical) for comparison to Aeolus observations as this increases the number of available data points significantly without introducing a distinct systemic error. For all flight legs performed during WindVal III and AVATARE, 56 % more data are available when applying the five-scanner-revolution average, keeping all the other parameters constant.

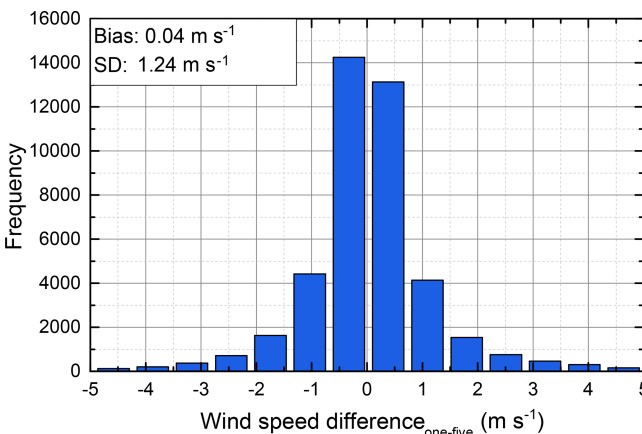

**Figure 3.** Histogram of the difference of wind speeds derived from 2 μm DWL data by means of one scanner revolution and five scanner revolutions ($v_{2\,\mu m1-scan} - v_{2\,\mu m5-scans}$) for all flights performed during the AVATARE campaign (see also Table 1). The mean and the standard deviation (SD) of the data are indicated by the inset.

### 4.3 Accuracy and precision of the retrieved wind speed

In order to assess the accuracy (systematic error) and precision (random error) of 2 μm DWL wind measurements, comparisons to dropsonde data were performed during several campaigns within the past years (Weissmann et al., 2005; Chouza et al., 2016; Reitebuch et al., 2017; Schäfler et al., 2018), and power spectra of LOS winds were analyzed (Witschas et al., 2017).

During the Gravity Wave Life-Cycle (GW-LCYCLE) I campaign (Wagner et al., 2017), the 2 μm DWL was used to measure horizontal and vertical wind speeds in order to investigate the life cycle of internal gravity waves induced by flow across the Scandinavian mountains. The spectral power of the vertical winds measured on a flight performed on 13 December 2013 at 5 km altitude indicates that the mean random error of LOS winds is $0.21\,\mathrm{m\,s^{-1}}$, and the mean systematic error of LOS winds is estimated to be smaller than $0.05\,\mathrm{m\,s^{-1}}$ (Witschas et al., 2017).

In addition, the random and systematic errors of 2 μm DWL wind speed measurements were determined by means of comparisons to dropsonde data. In particular, the data set acquired during the A-TreC campaign (Weissmann et al., 2005), the SALTRACE campaign (Chouza et al., 2016), the WindVal I campaign (Reitebuch et al., 2017) and the NAWDEX campaign (Schäfler et al., 2018) was used to determine the systematic error of retrieved wind speeds to be always below $0.1\,\mathrm{m\,s^{-1}}$ and the random error to vary between 0.92 and $1.5\,\mathrm{m\,s^{-1}}$. It is worth mentioning that both the systematic and the random errors are composed of the contribution of the 2 μm DWL and the dropsonde and corresponding representativeness errors. An overview of the respective results is given in Table 2.

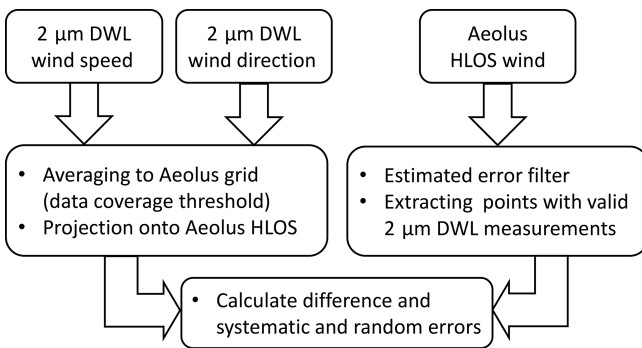

**Figure 4.** Sketch of the processing steps used to compare 2 μm DWL measurements with Aeolus observations.

The variability of the systematic and the random errors for different campaign data sets can have several causes, for instance slightly different thresholds for the allowed spatial and temporal distance between dropsonde and lidar observation and slightly different quality controls for the dropsonde and lidar measurements. Nevertheless, considering the low systematic error of smaller than $0.1\,\mathrm{m\,s^{-1}}$ and a reasonable random error varying between 0.92 and $1.5\,\mathrm{m\,s^{-1}}$, it can be concluded that the 2 μm DWL is a suitable reference instrument for Aeolus validation. For further analysis, the 2 μm DWL random error is considered to be $1\,\mathrm{m\,s^{-1}}$ for the horizontal wind speed.

### 5 Comparison of Aeolus and 2 μm Doppler wind lidar data

Due to the different horizontal and vertical resolutions of 2 μm DWL measurements ($\approx 8.4\,\mathrm{km}$, 100 m for one scanner revolution or $\approx 42\,\mathrm{km}$, 500 m) and Aeolus measurements ($\approx 90\,\mathrm{km}$ (Rayleigh) and down to $\approx 10\,\mathrm{km}$ (Mie), 0.25 to 2 km), averaging procedures are needed in order to compare respective wind observations. Furthermore, as Aeolus only provides HLOS winds, the 2 μm DWL measurements have to be projected onto the Aeolus HLOS direction. A sketch of the applied processing steps is shown in Fig. 4.

First, the wind speed and wind direction measured by the 2 μm DWL are averaged to the Aeolus grid by using the top and bottom altitudes as well as the start and stop latitudes given in the Aeolus Level 2B data product. As the 2 μm DWL does not provide full data coverage, a threshold for the number of available 2 μm DWL observations within an Aeolus grid point has to be set. In this study, at least 50 % valid 2 μm DWL measurements need to be available in order to consider the averaged wind speed and wind direction for further comparison. It was verified that using a more restrictive threshold of, for instance, 75 % or 90 % yields comparable systematic and random errors but with a significantly reduced number of data points that can be compared. Thus, it was decided to apply a threshold of 50 %.

**Table 2.** Systematic and random error of 2 µm DWL wind speeds determined by comparison to dropsonde measurements and power spectrum analysis of 2 µm DWL horizontal and LOS wind speeds.

| Wind product | Systematic error | Random error | Data points | Reference |
|---|---|---|---|---|
| Horizontal wind speed | $< 0.01\,\mathrm{m\,s^{-1}}$ | $1.20\,\mathrm{m\,s^{-1}}$ | 740 | Weissmann et al. (2005) |
| Horizontal wind speed | $0.08\,\mathrm{m\,s^{-1}}$ | $0.92\,\mathrm{m\,s^{-1}}$ | 1329 | Chouza et al. (2016) |
| Horizontal wind speed | $-0.03\,\mathrm{m\,s^{-1}}$ | $1.46\,\mathrm{m\,s^{-1}}$ | 938 | Reitebuch et al. (2017) |
| Horizontal wind speed | $0.05\,\mathrm{m\,s^{-1}}$ | $1.50\,\mathrm{m\,s^{-1}}$ | 245 | Schäfler et al. (2020) |
| Single LOS wind speed | $0.05\,\mathrm{m\,s^{-1}}$ | $0.20\,\mathrm{m\,s^{-1}}$ | 2000 | Witschas et al. (2017) |

Both the random error and the systematic error are composed of the contribution of the 2 µm DWL and the dropsondes, and corresponding representativeness errors.

Afterwards, all valid averaged wind speeds ($\mathrm{ws_{2\,\mu m}}$) and directions ($\mathrm{wd_{2\,\mu m}}$) are projected onto the horizontal LOS of Aeolus ($v_{2\,\mu m_{HLOS}}$) by means of the range-dependent azimuth angle $\varphi_{\mathrm{Aeolus}}$ that is provided in the Aeolus Level 2B data product according to

$$v_{2\,\mu m_{HLOS}} = \cos\left(\varphi_{\mathrm{Aeolus}} - \mathrm{wd_{2\,\mu m}}\right) \cdot \mathrm{ws_{2\,\mu m}}. \quad (1)$$

In a next step, the Aeolus HLOS winds (Rayleigh clear and Mie cloudy) are extracted for areas of valid 2 µm DWL measurements. Beforehand, the data are filtered by means of an estimated error for the wind speeds, which is also given in the Level 2B data product and which is estimated based on the measured signal levels as well as the temperature and pressure sensitivity of the Rayleigh channel response (Tan et al., 2008, 2017). In this study, a threshold for the estimated error of $8\,\mathrm{m\,s^{-1}}$ is applied for the Rayleigh winds and $4\,\mathrm{m\,s^{-1}}$ for the Mie winds.

The explained averaging procedure and the resulting data sets for the 2 µm DWL and Aeolus are illustrated in Fig. 5 for the satellite underflight performed on 17 November 2018. Panel (a) shows all valid Aeolus Rayleigh-clear observations, panel (b) shows the 2 µm DWL data averaged to the Aeolus measurement grid and projected onto its HLOS direction and panel (c) displays the corresponding Rayleigh-clear winds in regions where 2 µm DWL data are available. It can be seen that from 8719 available 2 µm DWL observations (see also Fig. 2), a comparison to only 72 Rayleigh-clear observations (13 Mie cloudy, not shown) is possible. Thus, a certain number of underflights are needed in order to obtain enough data points for a statistically significant comparison.

In order to validate the quality of Aeolus HLOS winds ($v_{\mathrm{Aeolus_{HLOS}}}$), the difference to the corresponding 2 µm DWL winds projected onto the Aeolus viewing direction ($v_{2\,\mu m_{HLOS}}$) is calculated according to

$$v_{\mathrm{diff_{HLOS}}} = v_{\mathrm{Aeolus_{HLOS}}} - v_{2\,\mu m_{HLOS}}. \quad (2)$$

$v_{\mathrm{diff_{HLOS}}}$ can also be used to verify the thresholds for the Aeolus estimated error used in this study as shown in Fig. 6. For the Rayleigh-clear winds (Fig. 6, top) it can be seen that the lowest estimated errors are calculated to be $3.7\,\mathrm{m\,s^{-1}}$.

The systematic error, represented by the difference of Aeolus and 2 µm DWL (Eq. 2), remains rather constant until an estimated error of about $8\,\mathrm{m\,s^{-1}}$ and then starts to increase gradually. The Mie-cloudy winds show estimated errors down to $0.7\,\mathrm{m\,s^{-1}}$. The systematic error is rather constant up to an estimated error value of $4\,\mathrm{m\,s^{-1}}$. For larger estimated errors, the systematic error increases remarkably. Thus, for further analysis, only Rayleigh-clear winds with estimated errors smaller than $8\,\mathrm{m\,s^{-1}}$ and Mie-cloudy winds with estimated errors smaller than $4\,\mathrm{m\,s^{-1}}$ are considered.

In order to quantify the quality of Aeolus HLOS winds, the bias and standard deviation (SD) of $v_{\mathrm{diff_{HLOS}}}$ are calculated by use of

$$\mathrm{bias} = \frac{1}{n} \sum_{i=1}^{n} v_{\mathrm{diff_{HLOS}}} \quad (3)$$

and

$$\mathrm{SD} = \sqrt{\frac{1}{n-1} \sum_{i=1}^{n} \left(v_{\mathrm{diff_{HLOS}}} - \mathrm{bias}\right)^2}, \quad (4)$$

where $n$ is the number of available data points. In addition to the standard deviation, the scaled median absolute deviation (scaled MAD) is calculated according to

$$\begin{aligned} &\mathrm{scaled\ MAD} \\ &= 1.4826 \times \mathrm{median}\left(\left|v_{\mathrm{diff_{HLOS}}} - \mathrm{median}\left(v_{\mathrm{diff_{HLOS}}}\right)\right|\right). \end{aligned} \quad (5)$$

The scaled MAD has the advantage that it is less sensitive to single outliers which may result in larger SD values and is thus used as a measure of the random error of Aeolus HLOS winds. The scaled MAD is identical to the standard deviation (Eq. 4) if the analyzed data are normally distributed. In addition to the aforementioned quantities, a least-square line fit to the respective data sets is performed, and the retrieved slopes and intercepts are evaluated.

All Aeolus wind results in relation to the averaged 2 µm DWL wind results for both the WindVal III and the AVATARE campaigns are shown in Fig. 7a and b, respectively, and are discussed in the next section.

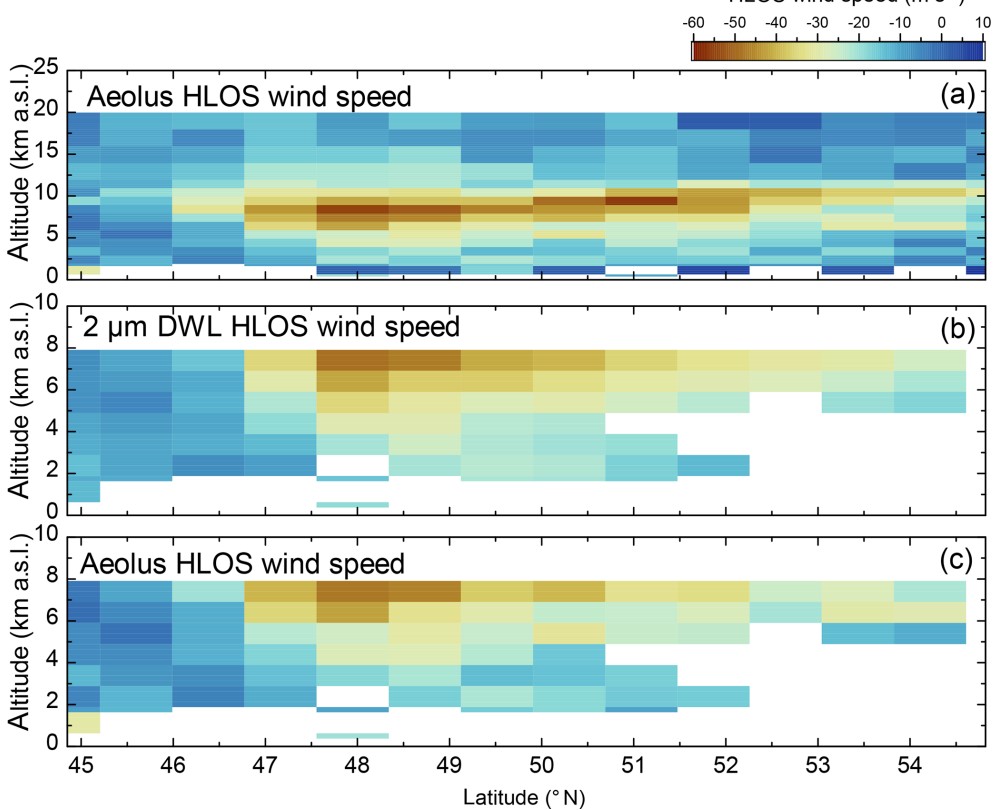

**Figure 5. (a)** Wind observations obtained during the first leg of the Aeolus underflight on 17 November 2018 between 45 and 55° N (1146 km) in the framework of the WindVal III campaign. **(a)** Aeolus Rayleigh-clear winds with an estimated error of smaller than 8 m s$^{-1}$. **(b)** Corresponding 2 μm DWL observations averaged to the Aeolus grid and projected onto its viewing direction. **(c)** Aeolus Rayleigh-clear winds as shown in **(a)** in regions where 2 μm DWL data are available for comparison.

## 6 Discussion

In Fig. 7, Rayleigh-clear winds and Mie-cloudy winds are indicated by blue dots and orange dots, respectively. Line fits to the corresponding data sets are depicted by the light blue and the yellow lines. The $x = y$ line is represented by the gray dashed line. A summary of the statistical parameters retrieved from the scatter plot analysis is given in Table 3.

All together, the four satellite underflights during the WindVal III campaign resulted in 231 data points for Rayleigh-clear wind validation and 109 data points for Mie-cloudy wind validation. The six satellite underflights during the AVATARE campaign resulted in 504 or 339 data points for Rayleigh and Mie wind validation, respectively, and thus about a factor of 2 more than for WindVal III. The increased number of data points can be explained by two more underflights performed during the AVATARE campaign and a better 2 μm DWL performance during AVATARE due to a complete optical realignment of the system before the campaign, leading to a remarkably better data coverage and hence to more data points being available for comparison. Additionally, since 5 March 2019 (08:44 UTC), Aeolus Mie

winds have been processed with a shorter horizontal averaging length of down to 10 km, also leading to more Mie winds that can be used for comparison. Furthermore, the range-gate settings of Aeolus were changed on 26 February 2019 (00:00 UTC) such that the vertical bins follow the ground elevation, which also increases the number of available data points.

The slope of the least-square line fits is close to 1 for both campaign data sets and both wind products (Mie cloudy and Rayleigh clear), indicating the good correspondence of the Aeolus HLOS wind data. No significant wind-speed-dependent bias is obvious from the slope analysis. In particular, the slope yields $0.99 \pm 0.01$ (Rayleigh) and $0.96 \pm 0.03$ (Mie) for the WindVal III data set and $0.98 \pm 0.02$ (Rayleigh) and $1.01 \pm 0.02$ (Mie) for the AVATARE data set. Here, the given uncertainty represents the standard error of the mean value retrieved from the least-square line fit. In the following, the magnitude of the systematic error and the random error retrieved from both campaign data sets and potential reasons for them are discussed.

**Table 3.** Comparison of Aeolus HLOS winds and 2 μm DWL winds projected onto the horizontal viewing direction of Aeolus.

|  | Rayleigh$_{clear}$ | | | | | Mie$_{cloudy}$ | | | | |
|---|---|---|---|---|---|---|---|---|---|---|
|  | Slope | Intercept | Bias | Scaled MAD | Points | Slope | Intercept | Bias | Scaled MAD | Points |
|  | (m s$^{-1}$)/(m s$^{-1}$) | (m s$^{-1}$) | | | | (m s$^{-1}$)/(m s$^{-1}$) | (m s$^{-1}$) | | | |
| WindVal III | $0.99 \pm 0.01$ | $2.2 \pm 0.3$ | 2.1 | 4.0 | 231 | $0.96 \pm 0.03$ | $2.7 \pm 0.4$ | 2.3 | 2.2 | 109 |
| AVATARE | $0.98 \pm 0.02$ | $-4.4 \pm 0.3$ | $-4.6$ | 4.4 | 504 | $1.01 \pm 0.02$ | $-0.21 \pm 0.17$ | $-0.17$ | 2.2 | 339 |

The uncertainty given for the slope and intercept values represents the standard error retrieved from the least-square line fit.

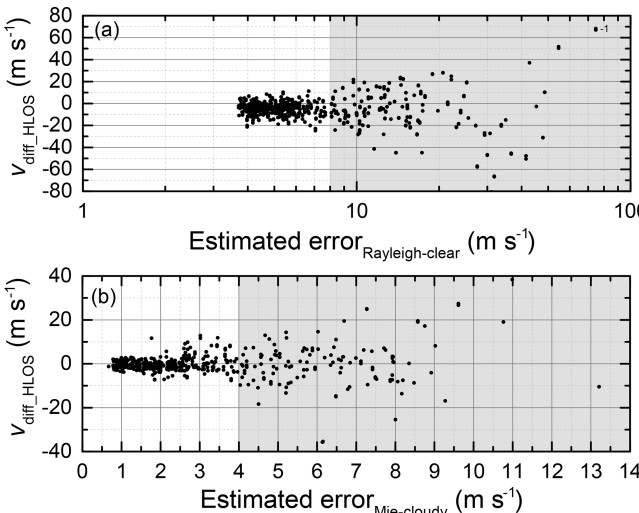

**Figure 6.** Wind speed difference of Aeolus HLOS winds and 2 μm DWL winds projected onto the Aeolus viewing direction according to Eq. (2) depending on the estimated error given in the L2B product for Rayleigh-clear winds **(a)** and Mie-cloudy winds **(b)**. Shown are all valid data points from WindVal III and AVATARE. Data points with an estimated error larger than 8 m s$^{-1}$ (Rayleigh) or 4 m s$^{-1}$ (Mie) are not considered to be valid observations (gray areas).

## 6.1 Systematic error

The intercepts of the respective line fits are determined to be $(2.2 \pm 0.3)$ m s$^{-1}$ (Rayleigh) and $(2.7 \pm 0.4)$ m s$^{-1}$ (Mie) for WindVal III and $(-4.4 \pm 0.3)$ m s$^{-1}$ (Rayleigh) and $(-0.21 \pm 0.17)$ m s$^{-1}$ (Mie) for AVATARE, where the uncertainty represents the standard error of the mean value retrieved from the least-square line fit. Except for the Mie winds of the AVATARE data, these values are considerably larger than the specified systematic error of 0.7 m s$^{-1}$ for Aeolus HLOS winds (ESA, 2016). A similar finding is obtained from the biases calculated according to Eq. (3) which yield 2.1 m s$^{-1}$ (Rayleigh) and 2.3 m s$^{-1}$ (Mie) for the WindVal III data set and $-4.6$ m s$^{-1}$ (Rayleigh) and $-0.17$ m s$^{-1}$ (Mie) for the AVATARE data set. As revealed in Sect. 4.3, the systematic error of 2 μm DWL observations is smaller than 0.1 m s$^{-1}$

and thus does not noticeably contribute here. Though the root cause of the enhanced systematic error is not unequivocally verified yet, it can be explained by an inadequate calibration file that is used within the Aeolus Level 2B processor, coupled with instrumental drifts that were observed throughout the mission time (Reitebuch et al., 2019). Such instrumental drifts require a regular update of the calibration file in order to avoid systematic errors in the wind retrieval which was not performed in the early stage of the mission.

It can also be seen that both the bias and the intercept of Rayleigh-clear winds change sign between the two campaigns, which is due to different calibration files used for the wind retrieval within the respective campaign periods. In particular, since the start of the mission on 22 August 2018, the very same calibration file was used until 16 May 2019 when a different calibration file was implemented. Thus, the Aeolus data acquired in the campaign period of WindVal III and AVATARE were processed with different calibration files, leading to the different systematic errors.

In order to further characterize and constrain the root cause of the enhanced systematic error, its dependency on several quantities, namely the time difference between 2 μm DWL and satellite observation, the actual wind speed, the scattering ratio, the altitude and the geolocation (latitude), is investigated, as shown in Fig. 8. The respective random error can be estimated by analyzing the spread of the systematic errors.

Due to the different platform speeds of the satellite ($\approx 7.7$ km s$^{-1}$) and the Falcon aircraft ($\approx 200$ m s$^{-1}$), almost all 2 μm DWL observations have a certain temporal difference with respect to the satellite observations. Depending on the variability of the atmospheric wind field, this can lead to both systematic and increased random errors for the comparison, where it is expected that both systematic and random errors increase with an increasing temporal difference between satellite and lidar observation. Thus, the obtained wind speed differences (Eq. 2) were analyzed depending on the time difference between satellite and 2 μm DWL observation as shown in Fig. 8a. In addition to the respective observation, the mean value of 50 observations and the corresponding standard deviation (error bars) are shown for the WindVal III data set (orange) and the AVATARE data set (magenta). It can be seen that data from about 1.5 h be-

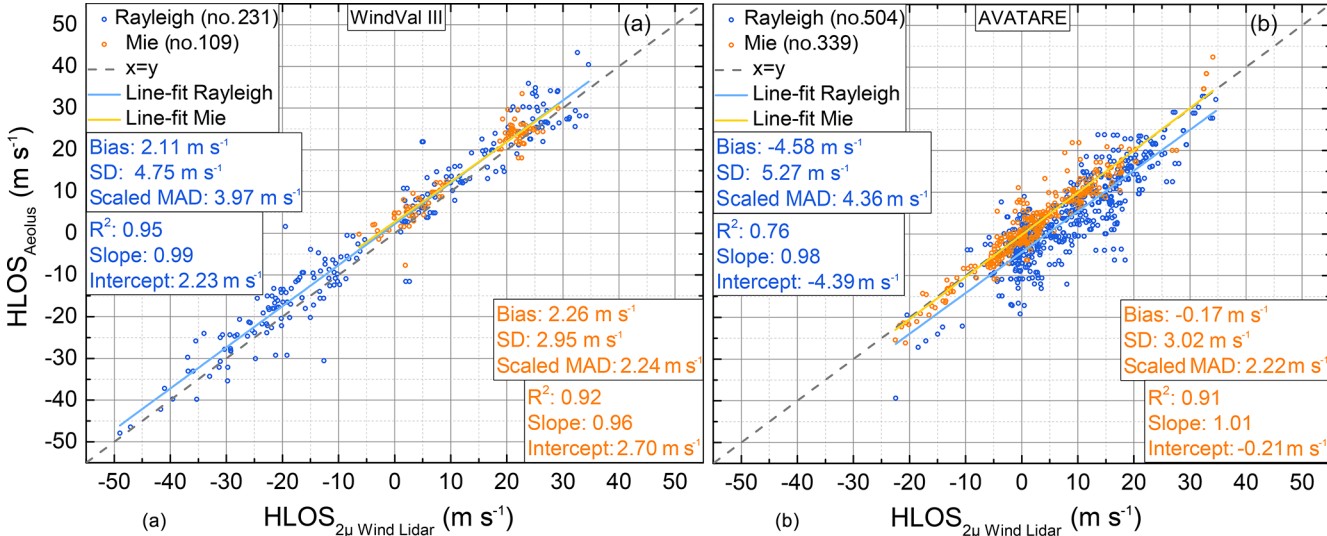

**Figure 7.** Aeolus HLOS wind speed plotted against the 2 µm DWL wind speed projected onto the horizontal viewing direction of Aeolus for eight flight legs from four underflights performed during the WindVal III campaign in 2018 **(a)** and for 10 flight legs from six underflights performed during the AVATARE campaign in 2019 **(b)** (see also Table 1). The wind measurements are separated in Rayleigh-clear winds (blue) and Mie-cloudy winds (orange). Corresponding least-square line fits are indicated by the light blue and yellow lines. The fit results are shown in the insets. The $x = y$ line is represented by the gray dashed line.

fore to 1.5 h after the satellite overflight are used for comparison. By analyzing the mean values and standard deviations, it becomes obvious that there is no significant increase in the systematic or the random error with an increasing time difference. Thus, a least-square line fit is performed for further analysis. The determined slopes of the respective data sets are $(1.1 \pm 0.4)\,(\mathrm{m\,s^{-1}})\,\mathrm{h^{-1}}$ for WindVal III and $(0.38 \pm 0.33)\,(\mathrm{m\,s^{-1}})\,\mathrm{h^{-1}}$ for AVATARE. Thus, a small linear trend with respect to the time difference of the satellite overflight is obvious from the WindVal III data set, whereas no significant dependency is obvious for AVATARE. The intercept values of $(2.2 \pm 0.3)\,\mathrm{m\,s^{-1}}$ and $(-4.7 \pm 0.2)\,\mathrm{m\,s^{-1}}$ are comparable to the mean bias obtained for the respective data sets, namely 2.1 and $-4.6\,\mathrm{m\,s^{-1}}$ (see also Fig. 7). Hence, it is verified that the time difference between satellite and 2 µm DWL observation does not introduce a significant systematic error for the statistical analysis of the data. It can also be seen that the points scatter randomly around the mean value with a comparable spread (see also error bars of mean values), indicating that the random error also does not have a remarkable dependency on the temporal difference of Aeolus and 2 µm DWL observations.

In the next step, the dependency of the systematic error of Rayleigh-clear winds on the actual wind speed represented by the 2 µm DWL measurements is investigated as shown in Fig. 8b. It can be seen that the acquired HLOS wind speed range was much larger for the WindVal III campaign (blue dots), ranging from $-50$ to $35\,\mathrm{m\,s^{-1}}$, whereas it was $-20$ to $35\,\mathrm{m\,s^{-1}}$ for AVATARE. Least-square line fits to the respective data sets yield a slope of $-0.014 \pm 0.015$ and

$-0.022 \pm 0.025$ and thus would indicate a wind speed dependency of the systematic error of about 1 % to 2 %. However, as the uncertainty of the slope has the same order of magnitude, this dependency is not considered to be significant. Additionally, the intercepts of $(2.2 \pm 0.3)$ and $(-4.4 \pm 0.3)\,\mathrm{m\,s^{-1}}$ are comparable to the mean bias obtained for the respective data sets stated above.

Another interesting topic to analyze is the dependency of the systematic error of Rayleigh-clear winds on the scattering ratio given in the L2B product as shown in Fig. 8c. It can be seen that there is a significant dependency of the systematic error on the scattering ratio for both campaign data sets. According to the least-square line fits applied to the respective data sets, the systematic error decreases from 3.4 to $1.0\,\mathrm{m\,s^{-1}}$ for WindVal III and from $-2.6$ to $-8.0\,\mathrm{m\,s^{-1}}$ for AVATARE within the available scattering ratio range. If one corrects this trend for the determined bias of $2.1\,\mathrm{m\,s^{-1}}$ (WindVal III) and $-4.6\,\mathrm{m\,s^{-1}}$ (AVATARE), the systematic error varies around zero, from $-1.1$ to $1.3\,\mathrm{m\,s^{-1}}$ (WindVal III) or from $-3.4$ to $2.0\,\mathrm{m\,s^{-1}}$ (AVATARE). Furthermore it can be seen that the scattering ratio varied between 1.08 and 1.18 for WindVal III and from 1.15 to 1.38 for AVATARE. This means that the determination of the scattering ratio, the respective threshold for classifying Rayleigh-clear winds or the actual aerosol load during the flights changed between the two campaigns. The slopes of the least-square line fits are determined to be $(-24.9 \pm 21.4)$ and $(-23.2 \pm 5.1)\,(\mathrm{m\,s^{-1}})^{-1}$ and thus even show similar magnitudes. The uncertainty of the obtained slope is smaller for the AVATARE data set as it extends over a broader scattering ratio range. A larger scattering ratio means

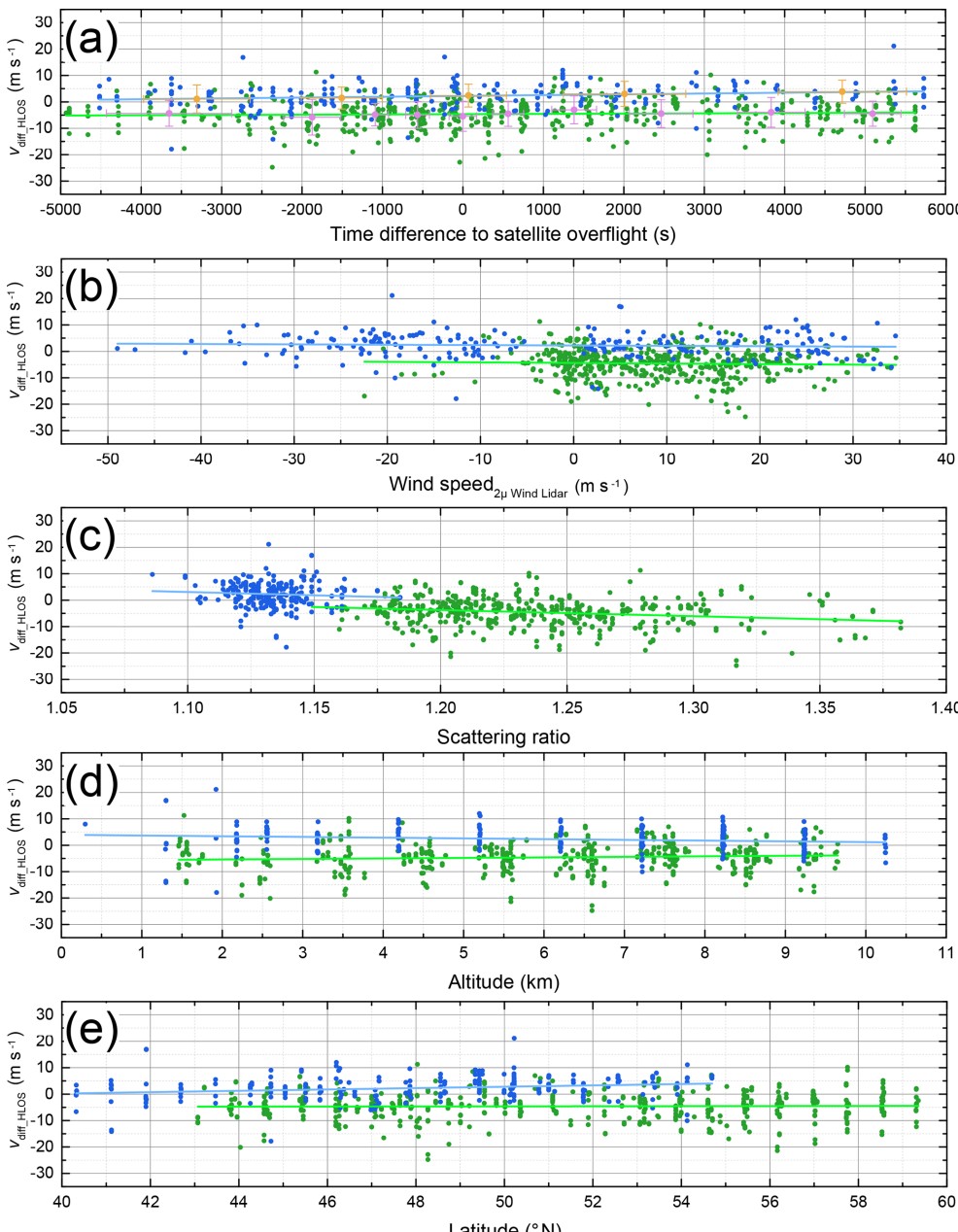

**Figure 8.** Wind speed difference of Aeolus Rayleigh-clear winds and 2 μm DWL winds calculated according to Eq. (2) depending on time difference of 2 μm DWL observation to satellite overflight time **(a)**, wind speed **(b)**, scattering ratio **(c)** and altitude **(d)** and latitude **(e)**. Data points of the WindVal III and AVATARE campaign are indicated in blue and green, respectively. Least-square line fits to the data points are represented by the light blue and light green lines. In plot **(a)**, the mean of 50 subsequent data points and the corresponding standard deviation (error bars) are show for WindVal III (orange) and AVATARE (magenta).

that there is a stronger contribution of the narrowband Mie return which also partly enters the Rayleigh spectrometer and hence results in a changed sensitivity of the Rayleigh channel. This has to be considered for the wind retrieval in order to avoid systematic errors. Hence, it is likely that this effect is not fully corrected so far, making the scattering ratio a significant contributor of the Rayleigh-clear wind systematic er-

ror. While writing this paper, improvements on the scattering ratio determination and correction scheme were already ongoing in the Level 2B processor (Jos de Kloe, personal communication, 7 August 2019).

The altitude dependency of the systematic error of Rayleigh-clear winds is shown in Fig. 8d. It can be seen that the Aeolus range-gate setting was kept constant during the

WindVal III campaign period (blue dots), leading to a vertical accumulation of wind observations. For the AVATARE campaign, the range gates followed the ground elevation, leading to a more scattered distribution of the data points. The least-square line fits to the respective data sets yield $(-0.27\pm0.12)$ and $(0.20\pm0.11)\,(\mathrm{m\,s^{-1}})\,\mathrm{km^{-1}}$ and thus indicate a small altitude dependency. Though it is not verified, this could be due to an imperfect temperature and pressure correction needed for the wind retrieval (Dabas et al., 2008) or an altitude-dependent scattering ratio during the flights. As two different calibration files were used for the Level 2B processing of Aeolus data within the respective campaign period, this could also explain the different slope sign for the two campaign data sets. However, more measurements would be needed in order to solidly determine if the systematic error shows a significant altitude dependency.

Lastly, the dependency of the systematic error of Rayleigh-clear winds on latitude is analyzed as indicated by Fig. 8e. It can be seen that 2 µm DWL observations are available from 40 to 60° N. The least-square line fits to the respective data sets yield $(0.26\pm0.08)\,\mathrm{m\,s^{-1}}$ per degree latitude north (WindVal III) and $(0.02\pm0.05)\,\mathrm{m\,s^{-1}}$ per degree latitude north (AVATARE). Thus, a small latitude dependency is obvious from the WindVal III comparison, but not for AVATARE. The analysis of Aeolus ground returns, which should actually yield $0\,\mathrm{m\,s^{-1}}$ wind velocity, has shown that there is a harmonic variation in the bias along the orbital phase (latitude dependence) (Reitebuch et al., 2019). In the future, this harmonic bias will be corrected by, for instance, exploiting ground return signals.

In summary, besides a generally incorrect calibration file, the scattering ratio or the corresponding correction scheme seems to be the main contributor to the systematic error of Rayleigh-clear winds. For Mie-cloudy winds the calibration file is considered to be the main reason for the enhanced systematic error. Given the small systematic bias of Mie-cloudy winds $(-0.17\,\mathrm{m\,s^{-1}})$ for the AVATARE campaign, it can be concluded that the strict requirement of $0.7\,\mathrm{m\,s^{-1}}$ specified for Aeolus HLOS winds can principally be met.

## 6.2 Random error

The random error $\sigma_{\mathrm{diff_{HLOS}}}$ given in Fig. 7 is represented by the scaled median absolute deviation according to Eqs. (4) and (5) and is determined to be $4.0\,\mathrm{m\,s^{-1}}$ (Rayleigh) and $2.2\,\mathrm{m\,s^{-1}}$ (Mie) for the WindVal III data set and $4.4\,\mathrm{m\,s^{-1}}$ (Rayleigh) and $2.2\,\mathrm{m\,s^{-1}}$ (Mie) for the AVATARE data set. As revealed in Sect. 4.3, the random error $\sigma_{\mathrm{DWL}}$ of 2 µm DWL observations lies between 0.92 and $1.50\,\mathrm{m\,s^{-1}}$. By assuming independence between Aeolus and 2 µm DWL measurements, the actual Aeolus random error $\sigma_{\mathrm{Aeolus}}$ can be calculated according to $\sigma_{\mathrm{Aeolus}} = \sqrt{\sigma_{\mathrm{diff_{HLOS}}}^2 - \sigma_{\mathrm{DWL}}^2}$, where $\sigma_{\mathrm{DWL}}$ is assumed to be $1\,\mathrm{m\,s^{-1}}$ here. With that, the random error of Aeolus HLOS winds is derived to be $3.9\,\mathrm{m\,s^{-1}}$

(Rayleigh) and $2.0\,\mathrm{m\,s^{-1}}$ (Mie) for the WindVal III data set and $4.3\,\mathrm{m\,s^{-1}}$ (Rayleigh) and $2.0\,\mathrm{m\,s^{-1}}$ (Mie) for the AVATARE data set. This demonstrates that the 2 µm DWL only contributes marginally to the random error and that the random error of Rayleigh-clear winds is significantly larger than the $2.5\,\mathrm{m\,s^{-1}}$ required for Aeolus HLOS winds at altitudes between 2 and 16 km (ESA, 2016; Kanitz et al., 2019; Reitebuch et al., 2019).

The main reason for the enhanced random error is a lower-than-expected signal level of the light backscattered from the atmosphere. On the one hand, this is caused by a lower laser pulse energy of about 53 mJ during WindVal III and 42 mJ during AVATARE instead of 80 mJ as originally planned for Aeolus (ESA, 2016; Kanitz et al., 2019; Reitebuch et al., 2019; Lux et al., 2020b). On the other hand, slight misalignments could introduce a clipping of the laser beam within the receiver at the field stop, leading to additional signal loss. Using a radiometric performance simulation tool, the detected signal levels are estimated to be a factor of 2.5 to 3 lower than expected (Reitebuch et al., 2019).

The 11 mJ decrease in laser pulse energy between the WindVal III and the AVATARE campaign periods also explains the increase in random error of the Rayleigh-clear winds from 4.0 to $4.4\,\mathrm{m\,s^{-1}}$. Considering that the random error is dominated by shot noise (Poisson noise), it is expected to scale with the square root of the laser energy. Thus, the expected random error for AVATARE can be calculated by considering the random error determined for WindVal III $(3.97\,\mathrm{m\,s^{-1}})$ and the respective mean laser energies (53 mJ for WindVal III and 42 mJ for AVATARE) according to $4.0\,\mathrm{m\,s^{-1}} \cdot \sqrt{53\,\mathrm{mJ}/42\,\mathrm{mJ}} = 4.5\,\mathrm{m\,s^{-1}}$, which is in good accordance with the determined random error of $4.4\,\mathrm{m\,s^{-1}}$, considering the uncertainties of the respective quantities. The Mie-cloudy wind random error does not show this trend, which is due to the fact that the Mie return signal depends not only on the laser energy but also on the presence of aerosols and clouds and their respective optical properties (backscatter and extinction coefficient) which can compensate for the lower laser power.

It is worth mentioning that all flights during WindVal III and AVATARE were performed under conditions where larger vertical wind speeds, induced by mountain waves for instance, can be excluded. The vertical winds measured by the 2 µm DWL confirm that the vertical wind speeds rarely exceed $0.5\,\mathrm{m\,s^{-1}}$. Thus, the vertical wind speed can be excluded as a distinct contributor to the Aeolus random error.

## 7 Summary

DLR recently performed two airborne campaigns with two wind lidars aboard DLR's Falcon aircraft over central Europe in November–December 2018 and June–July 2019 in order to validate ESA's Aeolus mission. A total of 10 satellite underflights with 19 flight legs covering more than 7500 km of

Aeolus swaths were performed and used to validate the preliminary wind data product of Aeolus by means of collocated observations for the first time. In this paper, the systematic and random errors of Aeolus HLOS wind observations are determined by means of the 2 µm DWL which acts as a reference system due to its low systematic and random errors that come along with the coherent measurement principle of the system (see Sect. 4.3). In particular, the systematic error of 2 µm DWL observations is smaller than $0.1\,\mathrm{m\,s^{-1}}$, and the random error is between 0.92 and $1.5\,\mathrm{m\,s^{-1}}$. Though this random error is noticeably smaller than that of Aeolus, it is considered for the statistical comparison performed here. The Aeolus measurement principle, its calibration procedures and wind data products are addressed in the context of an intercomparison study between Aeolus and A2D wind observations from the WindVal III campaign (Lux et al., 2020a).

For the WindVal III campaign, the systematic error is determined to be $2.1\,\mathrm{m\,s^{-1}}$ for Rayleigh-clear winds and $2.3\,\mathrm{m\,s^{-1}}$ for Mie-cloudy winds. For the AVATARE campaign, the systematic error is $-4.6\,\mathrm{m\,s^{-1}}$ (Rayleigh clear) and $-0.2\,\mathrm{m\,s^{-1}}$ (Mie cloudy). Except for the Mie-cloudy winds measured during the AVATARE campaign, the systematic error is remarkably larger than the $0.7\,\mathrm{m\,s^{-1}}$ value planned for Aeolus. Instrumental drifts together with inadequate calibration files are presumed to be the reasons for the enhanced systematic errors, which can and will be corrected in reprocessed data sets and which will be avoided for future data by improved algorithms.

Dependencies of the systematic error on observation time difference, wind speed, scattering ratio, altitude and geolocation were investigated, showing that the backscattering ratio has a remarkable influence on the systematic error of the Rayleigh-clear winds. This points to an issue with the crosstalk correction within the Level 2B retrieval which is currently revised.

It is worth mentioning that the Aeolus Level 2B product used in this study is still in an early stage and will also be improved based on the results of the airborne campaigns presented in this study. A few of the mentioned and discussed issues are already solved.

The random error of Rayleigh-clear winds is determined to be $4.0\,\mathrm{m\,s^{-1}}$ CE1 (WindVal III) and $4.36\,\mathrm{m\,s^{-1}}$ CE2 (AVATARE) and that of Mie-cloudy winds to be $2.2\,\mathrm{m\,s^{-1}}$ CE3 (WindVal III) and $2.2\,\mathrm{m\,s^{-1}}$ CE4 (AVATARE). Thus, for Rayleigh-clear winds, the random error is significantly larger than the $2.5\,\mathrm{m\,s^{-1}}$ planned for Aeolus HLOS winds at altitudes between 2 and 16 km. The enhanced random error is related to the lower laser energy together with an additional signal loss in the receiver possibly caused by clipping of the return signal on the field stop of the receiver. This also explains the even higher random error during the AVATARE campaign, where the mean laser energy was 11 mJ lower than during WindVal III.

The results elaborated in this study confirm the necessity to validate the Aeolus wind product and demonstrate that the

DLR airborne wind lidar payload is well suited for this task. In September 2019, another validation campaign is planned to be flown out of Keflavík, Iceland, in order to verify the performance of Aeolus in the North Atlantic region over a large wind speed range in the vicinity of the jet stream. This is also the first opportunity to investigate the performance of the second laser of Aeolus which has been operating since July 2019 during collocated airborne wind lidar observations.

*Data availability.* The 2 µm DWL data used in this paper can be provided upon request to the corresponding author of this paper (benjamin.witschas@dlr.de). Aeolus data were obtained from the VirES visualization tool, VirES for Aeolus (https://aeolus.services/, last access: 12 November 2019, ESA, 2019).

*Author contributions.* BW prepared the main part of the paper and performed the statistical comparison of 2 µm DWL and Aeolus data. CL was the principal investigator of both validation campaigns and supported both the analysis of lidar data and the preparation of this paper. AG supported the flight planning and performed the analysis of the meteorological conditions during the satellite underflights. OL provided an overview of all the satellite underflights performed during both campaigns. Additionally, he helped with the statistical analysis of comparing Aeolus and 2 µm DWL data. UM provided valuable insights into the Aeolus data processor and corresponding instrument calibrations that have a tremendous impact on the retrieved wind speeds. SR is the principal investigator of the 2 µm DWL, processed the 2 µm DWL data and contributed to the preparation of this paper. OR supported the analysis of satellite data and provided valuable comments on the respective processing steps used in the Aeolus processor. Additionally, he assisted in the preparation of this paper. FW provided the satellite data and performed corresponding analysis and contributed to preparing this paper.

*Competing interests.* The authors declare that they have no conflict of interest.

*Acknowledgements.* The technical assistance by Engelbert Nagel (DLR), Sammy Henderson and Dale Bruns (Beyond Photonics) as well as the support of the DLR flight facility for the realization of the performed validation campaigns is acknowledged, as are the valuable comments provided by Thomas Kanitz (ESA), Michael Rennie (ECMWF), Jos de Kloe (KNMI) and Thorsten Fehr (ESA). The authors acknowledge the provision of preliminary data (not fully calibrated and validated and not yet publicly released) of the Aeolus mission that is part of the European Space Agency (ESA) Earth Explorer program. Further data quality improvements, including in particular a significant product bias reduction, will be achieved before the public data release.

*Financial support.* The development of the ALADIN Airborne Demonstrator and the work carried out during the WindVal III campaign were supported by the German Aerospace Center (Deutsches

Zentrum für Luft- und Raumfahrt e.V., DLR) and the European Space Agency (ESA), providing funds related to the preparation of Aeolus (WindVal III (contract no. 4000114053/15/NL/FF/gp) and AVATARE (contract no. 4000128136/19/NL/ia)).

The article processing charges for this open-access publication were covered by a Research Centre of the Helmholtz Association.

*Review statement.* This paper was edited by Gerd Baumgarten and reviewed by three anonymous referees.

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

**Remarks from the language copy-editor**

CE1    Please give an explanation of why this needs to be changed. We have to ask the handling editor for approval. Thanks.

CE2    Please give an explanation of why this needs to be changed. We have to ask the handling editor for approval. Thanks.

CE3    Please give an explanation of why this needs to be changed. We have to ask the handling editor for approval. Thanks.

CE4    Please give an explanation of why this needs to be changed. We have to ask the handling editor for approval. Thanks.

**Remarks from the typesetter**

TS1    Please note that the corrections of "numbers" are not language changes. If you still insist on changing the two values in this table, the editor has to approve these changes.