# Peer review of "First validation of Aeolus wind observations by airborne Doppler Wind Lidar measurements"

_Atmospheric Measurement Techniques, 2019_

## Referee Comment (RC1) · Anonymous Referee #1 · 27 Jan 2020

The manuscript by Witschas et al. provides first results on the evaluation of the space-borne ALADIN wind lidar by comparing its measurements with those collected by an airborne coherent wind lidar. The comparison shows the results of two different validation campaigns and discusses the different possible sources for the observed differences. The paper is well organized and provides very valuable information for the lidar and atmospheric science communities. I recommend publishing on AMT after the following minor concerns are addressed:

1) My main concern is related with the technique used to retrieve the 2um lidar data. If I understand the procedure correctly, a sliding window on the LOS measurements is applied to increase the spatial coverage of the retrieval. I expect this sliding window (floating window in the paper) filter to introduce a spatial shift in the data which might

lead to an increase in the systematic error. A difference plot between the two retrievals shown in Fig. 2 (one scan vs five scans) might help to show the effect of this sliding window filter. For future evaluations, it might be worth using the ALADIN retrieval grid to group all the 2um lidar LOS measurement and retrieve 2um data natively on the ALADIN grid. It might be even possible to retrieve directly HLOS winds using the MFAS algorithm instead of retrieving first 3D winds and projecting them after into the ALADIN HLOS.

2) The authors use a threshold (8 m/s for Rayleigh and 4 m/s for Mie retrievals) based on the error reported in the L2B files to leave out from the evaluation some of the ALADIN retrievals. Do you know if during the assimilation of the ALADIN data by ECMWF similar filtering criteria are used? If that is the case, it would be good to use the same criteria for this study.

3) Although I expect the vertical component of the wind to have a small effect in the evaluation (considering the long spatial averaging), it might be worth mentioning it and maybe show an example of the retrieved 2um vertical component as a proof.

Specific comments:

1) Fig. 4: The Y axis scale could be reduced to -40/40.

2) Pag. 9, line 203: should be 'assess' instead of 'asses'

---

## Author Comment (AC1) · 19 Feb 2020

**(Author response)**

The manuscript by Witschas et al. provides first results on the evaluation of the spaceborne ALADIN wind lidar by comparing its measurements with those collected by an airborne coherent wind lidar. The comparison shows the results of two different validation campaigns and discusses the different possible sources for the observed differences. The paper is well organized and provides very valuable information for the lidar and atmospheric science communities. I recommend publishing on AMT after the following minor concerns are addressed:

1) My main concern is related with the technique used to retrieve the 2um lidar data. If I understand the procedure correctly, a sliding window on the LOS measurements is applied to increase the spatial coverage of the retrieval. I expect this sliding window (floating window in the paper) filter to introduce a spatial shift in the data which might lead to an increase in the systematic error. A difference plot between the two retrievals shown in Fig. 2 (one scan vs five scans) might help to show the effect of this sliding window filter. For future evaluations, it might be worth using the ALADIN retrieval grid to group all the 2um lidar LOS measurement and retrieve 2um data natively on the ALADIN grid. It might be even possible to retrieve directly HLOS winds using the MFAS algorithm instead of retrieving first 3D winds and projecting them after into the ALADIN HLOS.

Thanks a lot for this comment. As discussed on page 7, we use the MFAS algorithm in order to retrieve the wind vector from the raw data. The sliding window is thus not applied to single LOS wind speeds, but to the corresponding spectra of single LOS measurements. After accumulating the spectra, an FFT is performed in order to retrieve the wind. Using this procedure, it is not expected that a sliding window will introduce a "spatial shift" but rather smear out strong gradients. However, as the horizontal resolution of the Aeolus data (90 km for Rayleigh-clear winds) is more than twice of the sliding window size (~ 42 km), this effect is considered to be negligible.

In order to verify this hypothesis, the wind speed difference of 1-scan and 5-scan measurements from all AVATARE flights is analyzed for all common data points. This leads to the histogram shown in the figure below, which is also added to the paper manuscript. It can be seen that the systematic error of the wind speed difference is only **0.04 m/s**, which is negligible for Aeolus comparisons and which demonstrates that the sliding window does not introduce a distinct systematic error to the wind data.

Furthermore, the standard deviation of the wind speed difference is determined to be **1.24 m/s**. By assuming that both data sets (1-scan and 5-scan) contribute equally, the random error of 2-μm wind speed observations can be estimated to be **0.88 m/s**, which is in line with previous comparisons to dropsonde measurements as shown in section 4.3.

Still, for future evaluations it will be investigated if a spectral accumulation on Aeolus grid level will further improve our analysis or rather further increase the data coverage.

The manuscript was changed as follows (section 4.2):

In order to prove this hypothesis, the wind speeds retrieved by means of one scanner revolution ($v_{2\mu m_{1\text{-scan}}}$) and five scanner revolutions ($v_{2\mu m_{5\text{-scans}}}$) are analyzed. In particular, the difference of both data sets for all common data points of all flights flown during the AVATARE campaign (see also table 1) and the corresponding mean and standard deviation (STD) is calculated. An histogram of the wind speed difference is shown in Fig. 3. All together, more than 40000 data data points contribute to this analysis.

[Figure]

**Figure 3.** Histogram of the difference of wind speeds derived from 2-$\mu$m DWL data by means of one scanner revolution and five scanner revolutions (one minus five) for all flights performed during the AVATARE campaign (see also table 1). The mean and the standard deviation (STD) of the data are indicated by the inset.

It can be seen that the systematic error of the wind speed difference is 0.04 m/s and thus negligible for the comparison to Aeolus data. The random error (standard deviation) is determined to be 1.24 m/s. Assuming that both data sets contribute equally, the random error of 2-$\mu$m DWL wind speeds can be estimated to be $\sigma_{2\mu m} = (\sigma_{\text{difference}}/2)^{1/2} = 0.88$ m/s, which is in line with previous comparisons to dropsonde measurements as shown in section 4.3, table 2.

Considering that, it is decided to use the 2-$\mu$m DWL data retrieved by means of the modified MFAS algorithm using five scanner resolutions (horizontal) and five range gates (vertical) for comparison to Aeolus observations as it increases the number of available data points significantly without introducing a distinct systemic error. For all flight legs performed during WindVal III and AVATARE, 56% more data is available when applying the five-scanner-revolution average, keeping all the other parameters constant.

2) The authors use a threshold (8 m/s for Rayleigh and 4 m/s for Mie retrievals) based on the error reported in the L2B files to leave out from the evaluation some of the ALADIN retrievals. Do you know if during the assimilation of the ALADIN data by ECMWF similar filtering criteria are used? If that is the case, it would be good to use the same criteria for this study.

Yes, similar error thresholds using the estimated L2B errors are used by ECMWF before Aeolus data is used during the assimilation process (Rennie, M., L. Isaksen (2019): Guidance for Aeolus NWP Impact Experiments during the period September 2018 to November 2019, internal document available for registered Aeolus Cal/Val teams).

3) Although I expect the vertical component of the wind to have a small effect in the evaluation (considering the long spatial averaging), it might be worth mentioning it and maybe show an example of the retrieved 2um vertical component as a proof.

In general, all underflights performed during WindVal III and AVATARE were performed under conditions where larger vertical wind speeds, as for instance induced by mountain waves, can be excluded. This can also be seen by the vertical wind speed measurements from the 2-µm DWL, as exemplarily shown for the first flight of the WindVal III campaign (*Figure 2*). It can be seen that the vertical wind speeds are measured to lie between ± 0.5 m/s and are more or less varying randomly. Thus, the impact on the retrieved Aeolus winds is expected to be negligible.

[Figure]

*Figure 2. Vertical wind speed retrieved from 2-µm DWL data by means of the MFAS algorithm for one scanner revolution during the first ever Aeolus underflight performed on 17 November 2018 during the WindVal III campaign (corresponding horizontal wind speed is shown in Figure 1). White colors indicate areas with no valid wind measurements due to aerosol-poor atmospheric conditions and a corresponding insufficient SNR.*

During the most recent Aeolus validation campaign AVATARI performed from Iceland in September/October 2019, one particular research flight was performed over Greenland with predicted excitation of gravity waves. This flight is addressed to the investigation of the impact of larger vertical winds on the Aeolus wind product. The first analysis reveal vertical wind speeds of up to 2 m/s. The impact on the Aeolus winds is still under investigation. As the horizontal wavelength of these mountain waves is of the order of 10 km, the impact on Aeolus winds is still expected to be small as they average out for an Aeolus observation.

The manuscript was changed as follows (end of section 6.2):

It is worth mentioning that all flight during WindVal III and AVATARE were performed under conditions where larger
410    vertical wind speeds, as for instance induced by mountain waves, can be excluded. The vertical winds measured by the 2-µm DWL confirm that the vertical wind speeds rarely exceed 0.5 m/s. Thus, the vertical wind speed can be excluded as distinct contributer to the Aeolus random error.

Specific comments:

1) Fig. 4: The Y axis scale could be reduced to -40/40.

As the random error of Rayleigh-clear winds exceed +/- 40 m/s, only the scale of the Mie-cloudy error was adapted, as shown in the Figure below.

The figure is also adapted in the revised manuscript.

[Figure]

2) Pag. 9, line 203: should be 'assess' instead of 'asses'

Thanks a lot for this hint → corrected.

---

## Referee Comment (RC2) · Anonymous Referee #3 · 10 Mar 2020

[referee-annotated manuscript omitted]

---

## Author Comment (AC2) · 22 Mar 2020

**(Author response)**

The manuscript presents unique near simultaneous and near common volume observations of an airborne lidar and of the ALADIN instrument onboard the Aeolus satellite. The manuscript is well written and mostly well organized. The manuscript is a result of an extensive and valuable effort and deserves publication after some minor modifications. I recommend re-organizing Section 5, as the formulas used to calculate the data in figure 3 is given after the discussion of figure 4. So swapping Figure 3 and 4 and rearranging the text will allow the reader to follow the thoughts in a less distracting way. In line with this the figures 3 and 7 will be easier to understand if the y-axis label is the symbol defined in Eq. 2 ($v_{diffHLOS}$). Further comments are given in the supplement.

The Referee comments are extracted from the supplement and answered in the following.

p. 1, abstract: Time resolution, altitude resolution altitude range?
The specifications of the respective Aeolus data products are described in detail in section 3.2. Still, for the sake of completeness and in order to better interpret the random error, the following numbers were added to the abstract (altitude range, vertical resolution and horizontal resolution for Mie and Rayleigh winds). The time resolution is contained in the spatial resolution information.

3.9 m/s (Rayleigh) and 2.0 m/s (Mie) for WindVal III, and 4.3 m/s (Rayleigh) and 2.0 m/s (Mie) for AVATARE, whereas the Aeolus observations used here were acquired in an altitude range up to 10 km and have mainly a vertical resolution of 1 km (Rayleigh) and 0.5 km to 1.0 km (Mie) and a horizontal resolution of 90 km (Rayleigh) and down to 10 km (Mie).

p. 6, section 4: "operating at " or "operated by"
adapted to be "operated by".

p. 7, section 4.2: vertical resolution of 0.5 km?
the value was adapted to be 0.25 km

p. 8, "sliding window" should be added
In all places of the text, it is added that the 5 revolution scan retrieval is based on a sliding window. However, the term "about" is skipped, as the sliding window is always five pixel large.

p. 8, Fig. 2: in Fig. 5 (a) it is written km asl. I guess "km asl." is also the unit for this plot.
The y-axes label was adapted to km asl. Additionally, the colors were adapted to HCL color bar.

p. 9, How does the 5 scan x 500 m sliding window and the averaging to the Aeolus grid affect (reduce) the random errors of the 2µm DWL given here?
A new section was added to the paper manuscript (based on the comments of Referee #1), comparing the common wind results of 1 and 5 scans. This comparison shows an equal random error for both

retrievals. Thus, it seems that the random error does not remarkably reduce as the wind results are not independent (sliding window) and as the representativity gets worse. On the other hand, this comparison shown that the 5 scan revolution winds are at least not "worse" than the 1-scan ones. A dropsonde comparison is so far not available for the 5-revolution retrieval in order to perform an independent verification of the respective errors. The section added to the manuscript reads as follows:

In order to prove this hypothesis, the wind speeds retrieved by means of one scanner revolution ($v_{2\mu m_{1\text{-scan}}}$) and five scanner revolutions ($v_{2\mu m_{5\text{-scan}}}$) are analyzed. In particular, the difference of both data sets for all common data points of all flights flown

200 during the AVATARE campaign (see also table 1) and the corresponding mean and standard deviation (STD) is calculated. An histogram of the wind speed difference is shown in Fig. 3. All together, more than 40000 data data points contribute to this analysis.

[Figure]

**Figure 3.** Histogram of the difference of wind speeds derived from 2-$\mu$m DWL data by means of one scanner revolution and five scanner revolutions (one minus five) for all flights performed during the AVATARE campaign (see also table 1). The mean and the standard deviation (STD) of the data are indicated by the inset.

It can be seen that the systematic error of the wind speed difference is 0.04 m/s and thus negligible for the comparison to Aeolus data. The random error (standard deviation) is determined to be 1.24 m/s. Assuming that both data sets contribute

205 equally, the random error of 2-$\mu$m DWL wind speeds can be estimated to be $\sigma_{2\mu m} = (\sigma_{\text{difference}}/2)^{1/2} = 0.88$ m/s, which is in line with previous comparisons to dropsonde measurements as shown in section 4.3, table 2.

p. 10, table 2: 0.004 or <0.01
Value was changed to be <0.01 m/s

p.11, y-axes should be "v_diff_HLOS" instead of "error". Also in the text.
Changed for figure caption and in the text with additionally referring to the respective equation 2.

p.11, Fig.4: Is this for 17 November 2018 or for all data?
The graph shows all data points from WindVal III and AVATARE. The figure and the figure caption was modified accordingly.

[Figure]

**Figure 6.** Wind speed difference of Aeolus HLOS winds and 2-$\mu$m DWL winds projected onto Aeolus viewing direction according to Eq. (2) depending on the estimated error given in the L2B product for Rayleigh winds (top) and Mie winds (bottom). Shown are all valid data points from WindVal III and AVATARE. Data points with an estimated error larger than 8 m/s (Rayleigh) or rather 4 m/s (Mie) are not considered as valid observations (gray areas).

p. 12, Fig. 5: The data shown in subfigure (a) is already plotted in Fig. 2. It would make the interpretation of Figure 5 easier if it is not repeated here.
The upper panel (a) was removed from the figure in order to avoid any confusion, and the caption was adapted accordingly. Furthermore, the color map was adapted to HCL colors:

[Figure]

**Figure 5.** (a): Wind observations obtained during the first leg of the Aeolus underflight on 17 November 2018 between 45°N and 55°N (1146 km) in the framework of the WindVal III campaign. (a): Aeolus Rayleigh-clear winds with an estimated error of smaller than 8 m/s. (b): Corresponding 2-$\mu$m DWL observations averaged to the Aeolus grid and projected onto its viewing direction. (c): Aeolus Rayleigh-clear winds as shown in (a) in regions where 2-$\mu$m DWL data is available for comparison.

p. 12, Fig. 4 and Fig. 5: swap Figures 4 and 5 and re-arrange the corresponding discussion in the text
The figures were changed, and the text was adapted accordingly (see page 11 to page 14 in the new version of the manuscript). The content remained the same.

p. 13, Fig. 6: adapted color in label
done.

p. 14, Discussion: Does this increase the number of available data points or just diversifies their altitudes?
Actually it does both, it increases the number of available data points for comparison, by diversifying the altitudes due to smaller range gates in altitudes with airborne lidar measurements. The following sentence was added to the manuscript:

26 February 2019 (00:00 UTC) such that they follow the ground elevation which also increases the number of available data points due to smaller range gates in altitudes with airborne lidar measurements.

P. 15, Maybe add reference to Fig. 6 as a remainder.
Done

p. 17: Is there .a similar study (to Fig. 7) for Mie-cloudy winds? If yes and no correlations are found it would be worth mentioning here.
Currently, this study was performed for Rayleigh-clear winds only, as the number of available Mie-cloudy winds is not enough to perform a reasonable statistic. This is foreseen for the new campaigns data set of the AVATARI campaign (around Iceland and the North Atlantic), which provides more Mie-cloudy observations due to the modified Aeolus range gate setting applied during the campaign.

p. 18: Summary: It is stated three times in the manuscript that two wind lidars were onboard the Falcon aircraft. Only results from one lidar are shown. The reader wonders why. A short comment why only one lidar is used here may be useful.
The following paragraph was added for further clarification:

of Aeolus by means of collocated observations for the first time. In this paper, the systematic and random error of Aeolus wind observations is determined by means of the 2-$\mu$m DWL which acts as a reference system due to its low systematic and random error that comes along with the coherent measurement principle of the system (see also section 4.3). In particular, the

430  systematic error of 2-$\mu$m DWL observations is smaller 0.1 m/s and the random error is between 0.92 m/s and 1.5 m/s. Though this random error is noticeably smaller than the one of Aeolus, it is considered for the statistical comparison performed here. The Aeolus measurement principle, its calibration procedures and wind retrieval algorithms are addressed in the context of an intercomparison study between Aeolus and A2D wind observations from the WindVal III campaign (Lux et al., 2020).

p. 19, summary: The contribution by the 2μm DWL random error is not discussed here, probably because it is negligible. Please clarify.
Indeed, the 2μm-DWL random error contributes only marginally to the one of the scatter plot. Still, the random errors given for Aeolus were revised by considering a mean random error for 2-μm DWL observations of 1 m/s. The text was adapted accordingly.

**6.2 Random error**

The random error of the scatter plot $\sigma_{sc}$ represented by the scaled median absolute deviation according to Eq. (4) and shown

395  in Fig. 7 is determined to be 4.0 m/s (Rayleigh) and 2.2 m/s (Mie) for the WindVal III data set, and 4.4 m/s (Rayleigh) and 2.2 m/s (Mie) for the AVATARE data set. As revealed in section 4.3, the random error $\sigma_{DWL}$ of 2-$\mu$m DWL observations lies between 0.92 m/s and 1.50 m/s. By assuming independence between Aeolus and 2-$\mu$m DWL measurements the Aeolus random error $\sigma_{Aeolus}$ can be calculated according to $\sigma_{Aeolus} = \sqrt{\sigma_{sc}^2 - \sigma_{DWL}^2}$, where $\sigma_{DWL}$ is assumed to be 1 m/s here. With that, the random error of Aeolus HLOS winds is derived to be, 3.9 m/s (Rayleigh) and 2.0 m/s (Mie) for the WindVal III data set, and

400  4.3 m/s (Rayleigh) and 2.0 m/s (Mie) for the AVATARE data set. This demonstrates on the one hand, that the 2-$\mu$m DWL only contributes marginally to the random error, and that the random error of Rayleigh-clear winds is significantly larger than the 2.5 m/s required for Aeolus HLOS winds in altitudes between 2 km and 16 km (ESA, 2016; Kanitz et al., 2019; Reitebuch et al., 2019).